# The Hidden Role of Hydrogen Sulfide Metabolism in Cancer

**DOI:** 10.3390/ijms22126562

**Published:** 2021-06-18

**Authors:** Rong-Hsuan Wang, Yu-Hsin Chu, Kai-Ti Lin

**Affiliations:** 1Institute of Biotechnology, College of Life Science, National Tsing Hua University, Hsinchu 300, Taiwan; rong_hsuan_wang@gapp.nthu.edu.tw (R.-H.W.); rsp9840827@gmail.com (Y.-H.C.); 2Department of Life Science, College of Life Science, National Tsing Hua University, Hsinchu 300, Taiwan; 3Department of Medical Science, College of Life Science, National Tsing Hua University, Hsinchu 300, Taiwan

**Keywords:** hydrogen sulfide, gasotransmitter, persulfidation, cystathionine β-synthase, cystathionine γ-lyase, 3-mercaptopyruvate sulfurtransferase, cancer metabolism

## Abstract

Hydrogen Sulfide (H_2_S), an endogenously produced gasotransmitter, is involved in various important physiological and disease conditions, including vasodilation, stimulation of cellular bioenergetics, anti-inflammation, and pro-angiogenesis. In cancer, aberrant up-regulation of H_2_S-producing enzymes is frequently observed in different cancer types. The recognition that tumor-derived H_2_S plays various roles during cancer development reveals opportunities to target H_2_S-mediated signaling pathways in cancer therapy. In this review, we will focus on the mechanism of H_2_S-mediated protein persulfidation and the detailed information about the dysregulation of H_2_S-producing enzymes and metabolism in different cancer types. We will also provide an update on mechanisms of H_2_S-mediated cancer progression and summarize current options to modulate H_2_S production for cancer therapy.

## 1. Introduction

Hydrogen sulfide (H_2_S), a colorless, flammable, water-soluble gas, is recognized as the third gasotransmitter in 2002 [1]. Similar to the other two gasotransmitters, nitric oxide (NO) or carbon monoxide (CO), H_2_S acts as a critical mediator in multiple physiological processes, including regulation of blood vessel vasodilation [2,3,4], cardiac response to ischemia/reperfusion injury [5], and inflammation [6]. In mammalian cells, H_2_S is actively synthesized endogenously by three enzymes: cystathionine β-synthase (CBS), cystathionine γ-lyase (CTH), and 3-mercaptopyruvate sulfurtransferase (3-MST) [1,7]. Accumulated evidence indicates that dysregulation of these H_2_S producing enzymes was observed in multiple cancer types (See Section 4), suggesting H_2_S may play an important role during cancer development. Therefore, in this review, we will summarize the current understanding of H_2_S production, regulation, and biological functions during cancer development. We will particularly focus on how H_2_S-mediated protein persulfidation accomplishes cancer formation in different aspects of cancer hallmarks.

## 2. Hydrogen Sulfide

H_2_S is a colorless gas that smells like rotten eggs at low concentrations. Bernardino Ramazzini, the father of occupational health, wrote De Morbis Artificum Diatriba [8] (Diseases of Workers) in 1700. He described the effects of sewer gas exposure on the sewer workers, which causes irritation and inflammation to their eyes. In the early 19th century, people found out the major cause was H_2_S appearance in sewers that caused these harmful effects [9]. From then on, numerous toxicological effects of H_2_S on animals including humans have been reported [10,11]. Interestingly, although H_2_S was well-known as an environmental toxin, it can also be produced endogenously in bacteria [12], plants [13,14], and animals [15,16]. However, endogenously produced H_2_S was considered as a metabolic waste for a long time until K Abe and H Kimura suggested that the endogenous H_2_S functions as a neuromodulator in the brain in 1996 [17]. A few years later, Rui Wang proposed that H_2_S serves as the third gasotransmitter [1], while the first is NO and the second is CO. Gasotransmitters are endogenously produced small gaseous molecules and play different roles in multiple physiological conditions [18,19]. As the third gasotransmitter, H_2_S modulates a wide range of physiological processes, including smooth muscle relaxation [20,21], vasorelaxation [4], regulation of myocardial ischemia-reperfusion injury [22,23,24], neuron protection [25,26,27], inflammation [6,28,29], and angiogenesis [30,31].

## 3. Hydrogen Sulfide Mediated Protein Persulfidation

H_2_S regulates diverse cellular signaling pathways through persulfidation (S-sulfhydration) [3,32,33,34,35,36,37,38,39,40,41,42,43,44,45]. This novel and reversible posttranslational modification covalently adds a thiol group (-SH) to active cysteine residue (PSH/PS^-^) in its target protein, which forms protein persulfidation (PSSH/PSS^-^) (Figure 1A) [46]. The direct reaction between cysteine residue on the protein and hydrogen sulfide is unfeasible because of the thermodynamic constrains resulted from the release of hydrogen gas. Zhang et al. addressed several potential pathways for persulfidation under physiological conditions (Figure 1B,C) [47], including the reaction of S-Sulfenylated (PSOH) [48] or S-Nitrosated (PSNO) [49] proteins to form S-sulfhydrated modification through H_2_S [47]. In addition, protein disulfides can be reduced by H_2_S to form S-sulfhydrated proteins [50]. However, incubation of disulfide-containing protein, such as BSA [47] or immunoglobulin [51], with H_2_S did not lead to any detectable protein persulfidation. H_2_S can also be oxidized by metal centers such as heme iron and converted to HS• radical [52], which subsequently reacts with protein thiol and O_2_ to give protein persulfides in the end [47]. Furthermore, Greiner et al. confirmed the presence of polysulfides in NaHS solution and suggested that perhaps polysulfides are the actual persulfidating mediators than H_2_S (Figure 1C) [53]. Moreover, glutathione persulfide (GSSH/GSS^-^) and cysteine persulfide (CysSSH/ CysSS^-^), both are highly presented in mammalian cells and tissues, also thought of as possible persulfidating agents (Figure 1C) [54].

## 4. Regulation of Hydrogen Sulfide Production in Cancer

H_2_S has been admitted as a regulator of tumor progression and metastasis in recent years [55]. Endogenous H_2_S is catalyzed by three different H_2_S-producing enzymes, CBS, CTH, and 3-MST (Figure 2) [1,7]. Dysregulation of H_2_S-producing enzymes has been discovered in many cancer types (Summarized in Table 1). By regulating the expression of H_2_S-producing enzymes, the amount of tumor-derived H_2_S is changed, thereby altering the tumor microenvironment and affecting tumor growth and metastasis [56]. Therefore, in this section, we will summarize recent findings to unveil the possible regulatory mechanisms to modulate H_2_S production during cancer development.

### 4.1. CBS

CBS, which catalyzes H_2_S by driving beta-replacement, has been observed to be selectively upregulated in colon cancer, ovarian cancer, breast cancer, thyroid cancer, and gallbladder adenocarcinoma tissues [57,60,61,73]. CBS is a constitutively expressed enzyme and its activity can be regulated post-translationally [74]. The first reported post-translational modification of CBS is the small ubiquitin-like modifier (SUMO) modification [75]. SUMOylation facilitates CBS to translocate into the nucleus and further losses its catalytic activity [76]. Other than SUMOylation, CBS can be S-glutathionylated and then phosphorylated under oxidative stress, resulting in the increased activity of CBS and subsequent H_2_S production [77,78]. The catalytic activity of CBS can also be inhibited by the other two gasotransmitters, CO and NO, through binding to the ferrous heme of CBS [79]. Tu et al. observed that the DNA methylation on the CpG island of CBS promoter facilitates cell proliferation in colon cancer [80]. The activity of CBS could be allosterically elevated by S-adenosylmethionine (SAM), a universal methyl donor, which stabilizes CBS [81,82], to promote cell proliferation in colon cancer cells [83]. Additionally, CBS can also be controlled via its redox sensitivity through ^272^CXXC^275^ motif [84]. Under reductive-stress conditions, the redox-active disulfide bond (Cys^272^-Cys^275^) harbored by the CXXC motif induces the activity of CBS and further amplifies H_2_S production [84]. In contrast to the numerous studies in which CBS overexpression stimulates tumor growth in different cancer types, decreased CBS levels were also observed in glioma tumor cells, gastrointestinal cancer cells, and hepatocellular carcinoma [62,63,85]. The underlying mechanism remains unclarified, and reduced expression of CBS in glioma tumor cells may cause upregulation of 3-MST to generate H_2_S production alternatively [86].

### 4.2. CTH

CTH, another H_2_S-producing enzyme, is demonstrated as being up-regulated in several different cancer types, including prostate cancer, gastric cancer, and melanoma cells [45,66,87]. CTH is highly expressed in the liver, kidney, and brain [74]. Unlike CBS, CTH is an inducible protein stimulated by oxidative stress, ER stress, Golgi stress, inflammation, and starvation [88]. Expression of CTH is primarily controlled at the transcriptional level in response to cellular stress [74]. Nuclear factor (erythroid-derived 2)-like 2 (Nrf2) is a transcription factor responsible for antioxidant stress [89]. Under oxidative stress, Nrf2 induces CTH expression through binding to its antioxidant response element (ARE) at 5′-untranslated region (UTR) [90], resulting in the increased level of H_2_S production, and in turn, H_2_S stimulates Nrf2 expression as positive feedback [90]. Overexpression of another transcription factor, specificity protein (SP) 1, also modulates H_2_S generation through binding and activating to the core promoter of CTH [91]. Tumor necrosis factor α (TNFα) promotes H_2_S production through this SP1 mediated CTH expression pathway [33]. In prostate cancer, overexpression of CTH increased H_2_S production leads to the activation of nuclear factor-κB (NF-κB)-mediated interleukin 1β (IL-1β) signaling, resulting in the enhanced cell invasion, angiogenesis, lymphangiogenesis, tumor growth, and metastasis in prostate cancer [45]. In addition, induction of CTH expression by signal transducer and activator of transcription 3 (STAT3) signaling facilitates cell proliferation and migration in breast cancer, whereas induction of CTH expression by Wnt/β-catenin pathway stimulates cell proliferation in colon cancer [64,69]. CTH is also involved in the hepatoma cell proliferation via phosphorylation of extracellular signal-regulated protein kinase 1/2 (ERK1/2) through H_2_S [68].

### 4.3. 3-MST

3-MST is the only pyridoxal 5′-phosphate (PLP)-independent H_2_S-producing enzyme [74]. Unlike CBS and CTH, the catalytic activity of 3-MST is primarily regulated through its redox-sensitive characteristics [74], in which 3-MST is activated via oxidation at Cys^247^, the catalytically active site of 3-MST [92,93]. Although up-regulation of 3-MST in different cancer tissues has been confirmed, the underlying mechanism of 3-MST mediated H_2_S signaling is rarely discussed before [87,94]. Recently, several 3-MST inhibitors have been developed and the function of 3-MST in cancer can now be studied through inhibition of 3-MST activity [94,95]. More investigations will be needed to understand the underlying mechanism of 3-MST to evaluate the therapeutic potential of 3-MST specific inhibitors.

### 4.4. Hypoxia-Induced H_2_S Production

In cancer, hypoxia is a common feature of the microenvironment in solid tumors [96]. It is important to note that hypoxia profoundly evaluates the level of H_2_S because it inhibits the catabolism of H_2_S [97] and induces the expression of CTH [98]. Although the number of mitochondria decreases in cancer cells, mitochondria in cancer cells exhibit maximal sulfide-detoxifying capacity and a high level of sulfide:quinone oxidoreductase (SQR), which helps to transfer the H_2_S-derived electrons to the coenzyme Q (CoQ) [99]. The expression of H_2_S-producing enzymes and their translocation into mitochondria is enhanced under hypoxia, and subsequently increases the level of H_2_S [56,98]. In addition, H_2_S can stimulate ischemia-induced angiogenesis by enhancing the expression of hypoxia-inducible factor 1-alpha (HIF-1α) [73]. Zhou et al. revealed that H_2_S downregulated the expression of miR-640 and enhanced the expression of HIF-1α through the VEGFR2/mTOR pathway [100]. Wang et al. suggested that H_2_S might mediate HIF-1α via the PI3K/AKT pathway and promote the expression of vascular endothelial growth factor (VEGF) in non-small cell lung cancer [101]. In conclusion, cancer cells under hypoxia may produce H_2_S through induction of CTH to facilitate angiogenesis [102] and tumor growth.

## 5. The Role of Hydrogen Sulfide in Cancer

Dysregulation of H_2_S-producing enzymes was observed in multiple cancer types and hypoxia conditions as mentioned in the previous section, resulting in the increased level of endogenous H_2_S, thus contributing to cancer development in different aspects. In this section, we will focus on how H_2_S contributes to cancer progression through targeting different cancer hallmarks, including anti-apoptosis, DNA repair, tumor growth, cancer metabolism, metastasis, and angiogenesis (Summarized in Figure 3).

### 5.1. Hydrogen Sulfide in Anti-Apoptosis

Apoptosis is a naturally occurred and programmed cell death process in physiological and pathological conditions [103]. Evading apoptosis, one of the hallmarks during cancer progression, allows cancer cells to survive under various stresses [104]. The anti-apoptosis role of H_2_S has been recognized in different disease models, such as cardiovascular diseases [105], ischemia-reperfusion injury [106], and multiple cancer types [107,108,109,110]. One of the potential mechanisms of H_2_S-mediated suppression of apoptosis is scavenging reactive oxygen species (ROS) and reactive nitrogen species (RNS) by exerting the activities of classic antioxidants, like GSH and Trx, leading to profound antioxidant protection in cells [74]. The other potential mechanism is the activation of anti-apoptotic pathways through H_2_S-linked persulfidation on NF-κB [33], Kelch-like ECH-associated protein 1 (Keap1) [34], and Mitogen-activated protein kinase kinase1 (MEK1) [36]. 

Activation of NF-κB signaling stimulates multiple anti-apoptotic genes, including X-linked inhibitor of apoptosis protein (XIAP), cellular Inhibitors of Apoptosis Proteins (cIAPs), and the B-cell lymphoma 2 gene (Bcl-2) [111]. Activation of NF-κB requires translocation of NF-κB to the nucleus [111]. Persulfidation of NF-κB p65 subunit at Cys^38^ promotes its nuclear translocation [45] and promoter binding to those anti-apoptotic genes [33], resulting in the suppression of cellular apoptosis pathways [24,33]. 

Keap1, another protein mediated by persulfidation, is an adaptor of the Keap1-Cul3-RBX1 E3 ligase complex, which targets Nrf2 to proteasomal degradation through polyubiquitination [112]. Nrf2 is a transcription factor that controls genes containing antioxidant response elements (AREs) in their regulatory regions to escape from apoptosis [112]. Through H_2_S-mediated persulfidation at Cys^151^, Keap1 can undergo a conformational change which leads to the dissociation of Nrf2 from the Keap1-Cul3-RBX1 E3 ligase complex, and subsequently, the free Nrf2 translocates into the nucleus to exert its role on apoptosis escape [24,34,38,42,90].

MEK1, also known as MAP2K1, is one of the classical MAP kinase families that control a wide range of different cellular activities [113]. Activation of ERK1/2 by MEK1 generally inhibits apoptosis through modulating expressions of apoptotic-related proteins, including Bad, Bim-EL, Caspase 9, MCL-1, and TNFR [113]. Persulfidation of MEK1 at Cys^341^ leads to the phosphorylation of ERK1/2 and translocation of ERK1/2 into the nucleus to stimulate ERK1/2 mediated downstream signals in human endothelial cells and fibroblasts [36]. However, currently there is no direct evidence proving whether expressions of those apoptotic related genes are enhanced upon persulfidation of MEK1, more studies will be needed to clarify the role of persulfidation of MEK1 in anti-apoptosis.

### 5.2. Hydrogen Sulfide in DNA Repair

Protein poly [ADP-ribose] polymerase 1 (PARP1) is a well-known sensor of DNA single or double strand breaks, and thus it can initiate DNA damage repair pathways [114]. PARP1 inhibitor has been developed to create synthetic lethality of DNA repair systems in BRCA mutated cancers [115]. The idea is by blocking DNA repair pathways through PARP1 inhibitor in BRCA mutated cancers, the DNA damage responses will initiate signaling pathways to promote cell-cycle checkpoint activation, thus apoptosis will be triggered to eliminate cancer cells efficiently [115]. A study on MEK1 persulfidation indicates that persulfidation on MEK1 at Cys^341^ leads to MEK1 phosphorylation and translocation into the nucleus to stimulate PARP-1 activation and DNA damage repair, protecting cells from senescence [36]. Therefore, the activation of PARP1 through H_2_S mediated signaling may help to promote damaged cancer cell survival during cancer development. 

In addition to stimulating DNA repair pathway in cell nucleus, H_2_S also helps mitochondrial DNA (mtDNA) repair through persulfidation on mt-specific DNA repair enzymes EXOG at Cys^76^ [116]. The stimulation of this mtDNA repair pathway by H_2_S thus results in the apoptotic resistance to the cancer standard chemotherapy.

### 5.3. Hydrogen Sulfide in Tumor Growth

Elevated H_2_S-producing enzymes have been observed in multiple cancer types [45,57,64,94], and depletion of CBS or CTH activities results in the suppression of tumor growth in colon cancer [57], lung cancer [116], prostate cancer [45], and breast cancer [117]. Activation of MEK1, which belongs to the classical MAPK kinase pathways, is synonymous with cell proliferation and tumor growth [113]. Therefore, it is highly possible that ERK1/2 activities, which can be stimulated by H_2_S-mediated persulfidation on MEK1 [36], are the key drivers to promote tumor growth in CBS or CTH overexpressing tumors.

### 5.4. Hydrogen Sulfide in Cancer Metabolism

Exogenous H_2_S has a long history as an environmental toxin through inhibition of mitochondrial Complex IV, leading to the suppression of mitochondrial electron transport and inhibits aerobic ATP generation [118]. In contrast, endogenously produced H_2_S acts differently in mitochondria and cell metabolism. In mitochondria, H_2_S acts as a metabolic substrate to stimulates the mitochondrial electron transport chain [119]. Mitochondria are the powerhouse of cells to generate ATP via oxidative phosphorylation (OXPHOS). H_2_S oxidation by SQR, the mitochondrial respiratory Complex II, leads electron transfer to coenzyme Q (CoQ), facilitating the aerobic respiratory ATP synthesis [119]. In addition to serve as a metabolic substrate in the mitochondrial electron transport chain, H_2_S also increases the catalytic activity of mitochondria ATP synthase through persulfidation at Cys^244^ and Cys^294^ on the α subunit of ATP synthase (ATP5A1) [120], which may result in the higher ATP production in mitochondria through aerobic respiration. In cancer, currently it is still unclear whether this H_2_S-mediated mitochondria ATP production contributes to cancer progression, and we may guess tumor cells may generate ATP through this pathway only when O_2_ supply is sufficient.

On the other hand, tumor cells require the acquisition of necessary nutrients from the poor environment and utilize these nutrients to maintain viability and build new biomass [121]. To support their high growth rates on proliferation, cancer cells preferentially convert glucose to lactate by aerobic glycolysis even in the presence of sufficient O_2_ [121]. This phenomenon is so called Warburg effect [122], in which cancer cells adapt glycolysis to use the intermediates of the glycolysis to synthesize lipids, fatty acids, and nucleotides required for uncontrolled cell proliferation. To do that, cancer cells utilize lactate dehydrogenase A (LDHA) to elevate the rate of glycolysis [123]. The enzyme activity of LDHA thus is considered as a therapeutic target for the suppression of tumor growth and distant metastasis in different cancer types [123]. H_2_S-mediated persulfidation of LDHA at Cys^163^ enhanced its enzymatic activity, leading to the increased production of lactate in HCT116 colon cancer cells [124]. Consistent with these observations, depletion of H_2_S production by CBS knockdown resulted in the reduced oxygen consumption and ATP production in both colon cancer [57] and ovarian cancer cells [58], indicating the importance of H_2_S in the modulation of cancer metabolism to support tumor cell uncontrolled growth.

### 5.5. Hydrogen Sulfide in Cancer Metastasis

Cancer metastasis is an important milestone during cancer development, in which cancer cells invade surrounding tissues, spread to distant sites, and grow secondary tumors in another part of the body [104]. The initial development of cancer metastasis requires cancer cells to gain migration and invasion ability through epithelial to mesenchymal transition (EMT) [125]. Endogenous H_2_S promotes cancer cell migration and invasion in multiple cancer types, such as prostate cancer [45], lung cancer [101], colon cancer [95,126], and liver cancer [109], partly through induction of ATP citrate lyase (ACLY) to facilitate EMT [95]. Moreover, NF-κB, a key molecule driving cancer metastasis [111], is involved in the H_2_S-modulated cancer metastasis through persulfidation. Persulfidation at Cys^38^ of the NF-κB p65 subunit facilitates nuclear translocation of p65 and then induces expressions of metastatic promoting genes, especially IL-1β, resulting in enhanced cell invasion and distant metastasis during prostate cancer progression [45]. 

### 5.6. Hydrogen Sulfide in Angiogenesis

Angiogenesis is the formation of new blood vessels from the pre-existing vasculature [127]. During cancer development, tumor cells secrete pro-angiogenic factors, such as VEGF, to support tumor growth and stimulate distant metastases [127]. Numerous studies already confirmed that H_2_S acts as a pro-angiogenic factor in vitro and in vivo under different physiological and disease conditions, including cancer [102]. Silencing H_2_S producing enzyme, CBS, reduces the formation of tumor blood vessels in colon cancer [17,57] and ovarian cancer [58]. Depletion of another H_2_S producing enzyme, CTH, not only blocks angiogenesis [30,45] but also lymphangiogenesis [45]. Moreover, H_2_S promotes hypoxia-induced angiogenesis through induction of HIF-1α as we previously discussed in Section 4.4.

Although H_2_S is an endogenous stimulator of angiogenesis through activation of PI3K/AKT and MAPK signaling pathway [30], the underlying mechanism remains unclear. One possibility is H_2_S may mediate angiogenesis through persulfidation of Kir6.1 subunit of KATP channel at Cys^43^ [3] since pharmacological inhibition of KATP channel attenuates VEGF mediated endothelial cell migration [30]. The other possibility is through H_2_S mediated persulfidation of NF-κB p65 subunit and subsequent activation of NF-κB/IL-1β signaling [45] since IL-1β is a known pro-angiogenic cytokine during cancer progression through induction of VEGF [57,128]. More research will be needed to decipher how H_2_S impacts angiogenesis during cancer development.

## 6. Hydrogen Sulfide Based Therapeutics

Upregulation of H_2_S producing enzymes and increased endogenous H_2_S production are recognized in many cancer types, which in turn promotes cancer progression. However, donors producing a higher level of H_2_S are considered as anti-cancer drugs [129] through induction of uncontrolled intracellular acidification [130], resulting in the promotion of apoptosis [131,132,133,134,135,136] and cell cycle arrest [131,133,137,138]. The controversial role of H_2_S in cancer research field can be explained by the bell-shaped (biphasic) model, in which Hellmich and Szabo suggested that lower concentrations of H_2_S display pro-cancer effects while higher concentrations exhibit anti-cancer properties [55,139]. In that sense, both H_2_S inhibitors and donors show some potential on cancer therapy.

The donors of H_2_S include sulfide salts, such as sodium hydrosulfide (NaHS) and sodium sulfide (Na_2_S), which release H_2_S directly. Other H_2_S donors are categorized by their release mechanisms. Donation of H_2_S can be triggered by hydrolysis, reactive oxygen species (ROS), biological thiols, specific wavelengths of light, and enzymes [140,141,142,143]. Various H_2_S donors have been synthesized and tested preclinically to kill cancer cells at high doses and/or long-term exposure [131,133,135,144,145,146,147]. A slow-releasing H_2_S donor, GYY4137, enhances glucose uptake, glycolysis, and lactate production while decreasing the activity of pH regulators, anion exchanger (AE), and sodium/proton exchanger (NHE), resulting in the intracellular acidification in cancer cells [130]. Moreover, GYY4137 blocks STAT3 signaling, leading to the cell cycle arrest, apoptosis, and inhibition of hepatocellular carcinoma tumor growth [133]. Other H_2_S donors, diallyl trisulfide (DATS) and 5-(4-hydroxyphenyl)-3H-1,2-dithiole-3-thione (ADT-OH), are also effective in the suppression of tumor growth through inhibition of NF-κB activity and upregulation of Fas-associated protein with death domain (FADD) in melanoma [134,136]. However, the toxicity of these H_2_S donors in normal cells are the major concerns for the current drug development.

In contrast to H_2_S donors, options to inhibit endogenous H_2_S production are very limited. Currently, there are only inhibitors for CTH and CBS, both are PLP-dependent enzymes. The most frequently used inhibitor, DL-propargylglycine (PAG) [148], is an irreversible inhibitor of CTH with IC50 at 40 μM and displays high selectivity for CTH over CBS [149]. However, PAG is typically used at millimolar concentrations, in cell-based assays [30,64,69,150,151,152], due to limited cell permeability [153]. Studies have confirmed that millimolar concentration of PAG results in the non-selective inhibition on enzymes other than CTH [154,155,156]. The other frequently used inhibitor is aminooxyacetic acid (AOAA), which inhibits both CBS and CTH and shows higher potency against CTH (IC50 at 1.1 μM) than CBS(IC50 at 8.5 μM) [149]. Additionally, AOAA is a general inhibitor of several other PLP-dependent enzymes [157] including cysteine aminotransferase (CAT), which catalyzes the transamination between L-cysteine and α-ketoglutarate (α-KG) to produce 3-mercaptopyruvate (3-MP), a substrate for 3-MST (Figure 2) [158]. Furthermore, AOAA inhibits non-enzymatic H_2_S production catalyzed by iron (Fe^3+^) and PLP [159]. As the result, AOAA suppresses H_2_S production through all enzymatic and non-enzymatic pathways. There are several additional molecules that selectively inhibit activities of CBS [160,161,162,163,164] or CTH [165,166,167,168,169] to exert anticancer effect. On the other hand, there was no selective inhibitor for 3-MST until recently, Hanaoka, K. et al. (2017) established a high-throughput screening (HTS) method to screen 174,118 compounds and several potential inhibitors for 3-MST were identified. Among them, 2-[(4-hydroxy-6-methylpyrimidin-2-yl)sulfanyl]-1-(naphthalen-1-yl)ethan-1-one (HMPSNE) showed the highest selectivity for 3-MST [170,171,172] and dose-dependently inhibited cell proliferation in colon cancer cell [94]. In a more recent study, derivative of HMPSNE was synthesized and exerted antiproliferative effect in vitro and in vivo in colon cancer model through targeting 3-MST [173]. More investigations will be needed to confirm their potency and efficacy in the inhibition of H_2_S production.

## 7. Conclusions

Given the fact that the importance of H_2_S-mediated persulfidation in protein functions, it is not surprising that aberrant expressions of H_2_S producing enzymes can contribute to cancer development from different aspects, including anti-apoptosis, DNA repair, tumor growth, cancer metabolism, metastasis, and angiogenesis (Figure 3). However, right now there are only very limited and non-specific options available for pharmacological inhibitors to suppress H_2_S production. For future perspectives, we hope more H_2_S-targeted signaling molecules will be identified and pharmacological inhibitors with high selectivity and potency will be developed to improve the future experimental therapy of cancer.

## Figures and Tables

**Figure 1 ijms-22-06562-f001:**
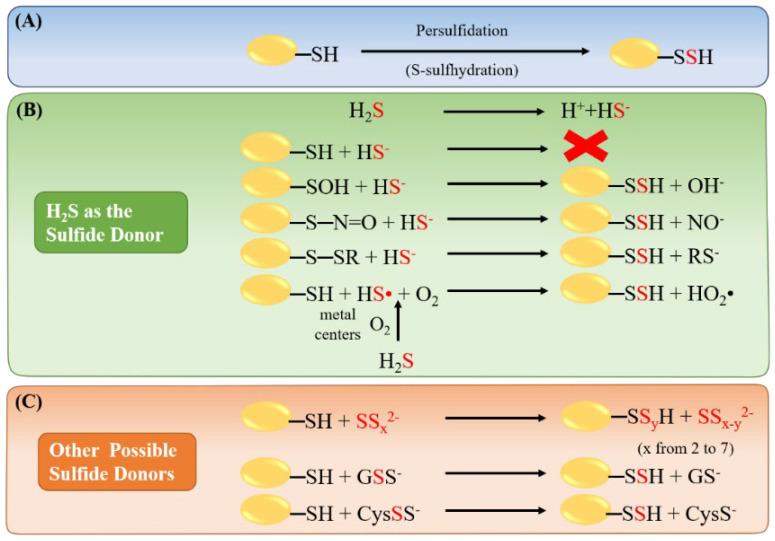
Possible reaction mechanisms for protein persulfidation. (**A**) Illustration of protein persulfidation, also called protein S-sulfhydration. (**B**) Proposed possible persulfidating reaction by H_2_S as the sulfide donor. In solution, H_2_S will dissociate into HS^−^ and H^+^. A direct reaction between protein thiol and HS^-^ is impossible. In contrast, persulfidation can result from a sulfide anion on an oxidized protein thiol, including S-OH, S-N=O, and S-SR. HS• radical can be generated by H_2_S through oxidation by metal centers. HS• will then react O_2_ to generate protein persulfidation and HO_2_•. (**C**) Other sulfide donors, such as polysulfides, glutathione persulfide (GSS^-^), and cysteine persulfide (CysSS^-^), may also act as persulfidating agents to stimulate protein persulfidation.

**Figure 2 ijms-22-06562-f002:**
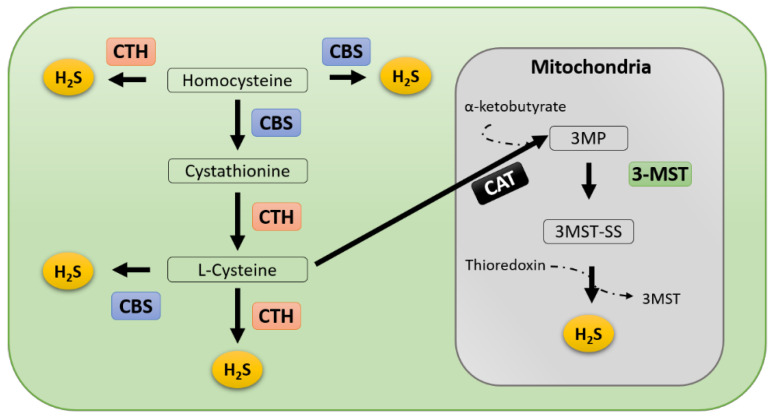
Simplified illustration of endogenous H_2_S production pathways. Three principal enzymes responsible for H_2_S production are CBS, CTH, and 3-MST. Homocysteine is the major substrate for H_2_S production. CTH and CBS generate H_2_S majorly in the cytosol, while 3-MST generate H_2_S in mitochondria. CBS, cystathionine β-synthase; CTH, cystathionine γ-lyase, 3-MST, 3-mercaptopyruvate sulfurtransferase; CAT, cysteine aminotransferase.

**Figure 3 ijms-22-06562-f003:**
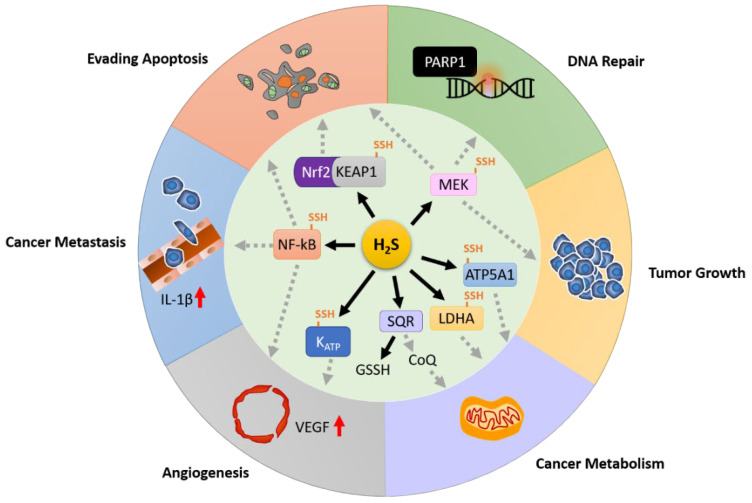
**The potential role of H_2_S during cancer development.** Illustration of the six cancer hallmarks mediated by H_2_S modulated protein persulfidation or using H_2_S as a metabolic substrate for electron transfer.

**Table 1 ijms-22-06562-t001:** Overview of upregulation and downregulation of three H_2_S producing enzymes in different cancer types.

H_2_S-Producing Enzymes	Dysregulation	Cancer Types
**CBS**	Upregulation	colon cancer [57]
		ovarian cancer [58]
		breast cancer [59]
		thyroid cancer [60]
		gallbladder adenocarcinoma [61]
	Downregulation	hepatocellular carcinoma [62]
		gastrointestinal cancer [63]
**CTH**	Upregulation	breast cancer [64]
		prostate cancer [65]
		gastric cancer [66]
		bladder cancer [67]
		hepatoma [68]
		colon cancer [69]
	Downregulation	clear cell renal cell carcinoma [70]
**3-MST**	Upregulation	glioma tumor [71]
		colon cancer [72]
	Downregulation	Unknown

## Data Availability

Not applicable.

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
