# Peer review of "The Hidden Role of Hydrogen Sulfide Metabolism in Cancer"

_ijms, 2021, doi:10.3390/ijms22126562_

Round 1

Reviewer 1 Report

MAJOR

Page 4: Please clarify what do you mean that CTH is not constitutively expressed; it is known to be constitutively expressed in many cells and tissues.

Page 4: “the underlying mechanism of 3-MST mediated H2S signaling remains unclear due to the lack of specific inhibitors” is not really correct. There are some inhibitors and some studies characterizing this process. In fact authors cite some of these papers later (Refs 90 and 130 for example. This part needs to be corrected)

Table 1 is not comprehensive. Either make it comprehensive or change it to “Examples” instead of “Summary” in its title

The part on H2S and DNA repair is incomplete. For example Ref 118 showing that H2S helps with the assembly of mitochondrial DNA repair complexes is not discussed.

Page 7: depletion of CBS or CTH activities results in the suppression of tumor 233 growth in prostate cancer [48], lung cancer [118], breast cancer [119], and colon cancer 234 [60]. Please arrange this sentence chronologically. The first paper published on the subject was reference 60.

“Mitochondria is the powerhouse of cells”.. “Mitochondria” is PLURAL!!! Mitochondria ARE, not mitochondria IS. (mitochondrion is singular, mitochondria is plural.)

Section 5.5. When discussing metastasis, you should also discuss invasion and the process of MET/EMT and teh recent studies implicating ACLY and H2S in this (Ref 130 for example)

Also the following report should be discussed here: Cancer Res. 2017 Nov 1;77(21):5741-5754.

Section 5.6. When discussing angiogenesis, you should at least mention the two 2013 studies (colon and ovarian cancer) showing that when the transplanted tumor has CBS silenced, the peritumor angiogenesis response is reduced. Also there are some more recent studies on this topic. This section seems under developed and needs expanding.

Page 8: “in  which Hellmich, Coletta et al. suggested that lower concentrations of H2S display pro-cancer effects while higher concentrations exhibit anti-cancer properties” This study has two senior authors, Hellmich and Szabo (not Coletta).

In that sense, both H2S inhibitors and donors show great promises on potential cancer therapy” (the “great promises” is a bit overstated. Just say “show some potential”

“Various H2S donors have been developed” – they have not been developed. They have  been synthesized and tested preclincially but not have been developed (“developed” would mean they are in clinical use)

Last sentence” will be developed to benefit more cancer patients” – instead of “more cancer patients” say “developed to improve the future experimental therapy of cancer”

MINOR

Some small editorial issues need to be corrected: for example in the keywords some items are capitalized (unnecessarily); others are not. Same in the text; no need to capitalize interleukin or nuclear factor kappa B or DL-propargylglycine, or aminooxyacetic acid  in the middle of sentences.

Some incorrect English usage, eg

“CTH also involves in the hepatoma cell proliferation”

Page 9: “On the other hand, there was no selective inhibitor for 3-MST until recently, Ha-330 naoka, K., et al. (2017) established a high-throughput screening (HTS) method to screen 331 174,118 compounds and several potential inhibitors for 3-MST were identified” – there is a more recent follow-up on this work, with in vivo demonstration of antitumor effect of a 3MST inhibitor. Please incorporate this into the discussion.

J Med Chem, . 2021 Apr 15.

Marina Bantzi  - Novel Aryl-Substituted Pyrimidones as Inhibitors of 3-Mercaptopyruvate Sulfurtransferase with Antiproliferative Efficacy in Colon Cancer

DOI: 10.1021/acs.jmedchem.1c00260

Also there are several additional CBS inhibitor approaches that have antitumor effects. These are well summarized in a table in the following review. Perhaps this review should be cited and these approaches should be expanded. Cystathionine-β-Synthase: Molecular Regulation and Pharmacological Inhibition. Zuhra K, Augsburger F, Majtan T, Szabo C. Biomolecules. 2020 Apr 30;10(5):697. (or each individual paper on CBS inhibitors’ antitumor effects should be incorporated; this would probably mean another 10 references).

Also the references must be checked; some are listed twice, for example 4 and 22 are the same (even though they are formatted slightly differently in their page numbers.. all of these formatting issues should be consistent)

Reviewer 2 Report

I the article titled “The hidden role of hydrogen sulfide metabolism in cancer” by Wang, R-H, et al., the authors provide a concise review of what is known regarding endogenous hydrogen sulfide production and its role in promoting or suppressing cancers. They also address exogenous addition of H2S and/or H2S donors as tools for anti-cancer therapies and approaches. Overall, the article is well written and summarizes relatively current trends in the field. I only have a few minor suggestions (listed below).

1) The authors go in depth into the pro-cancer mechanisms of H2S that appear to be cell/tissue type specific. However, they do not go into great detail in terms of the mechanisms that H2S can serve as a tumor suppressor.  It would be worthwhile, perhaps in the “Hydrogen Sulfide Based Therapeutics” section, to include this information.

2) Line 50: should read “Gasotransmitters are endogenously produced small gaseous molecules and play different roles in multiple physiological conditions”

3) Line 136: should read “overexpression of CTH increased H2S production and leads to”

4) Line 142: should read “CTH is also involved in hepatoma cell proliferation via”

5) Line 143: It is unclear why the authors made the statement at the end of the CTH section “Nevertheless, elevated H2S production or homocysteine level negatively regulated CTH expression while promoting CBS expression in cardiomyocytes” as this has nothing to do with the role of H2S in cancer. I suggest to remove this sentence from this section.

6) Line 172: I would suggest to re-organize Table 1 as it is difficult to follow what cancer types fall under what enzyme and what dysregulation direction. Perhaps the addition of lines to separate these groups would be useful.

7) Line 202: should read “which targets nuclear factor-erythroid –related factor-2 (Nrf2) to proteasomal degradation through polyubiquitination”

8) Line 238: should read “are the key drivers to promote tumor growth”

9) Line 255: should read “tumor cells require the acquisition of necessary nutrients from the poor environment”

10) Line 308: should read “The donors of H2S include sulfide salts, such as sodium hydrosulfide (NaHS), and sodium sulfide (Na2S), which release H2S directly.”

11) Line 327-328: It should be noted that AOAA also inhibits non-enzymatic production of H2S catalyzed by iron and pyridoxine (Yang, J. et al. Non-enzymatic hydrogen sulfide production from cysteine in blood is catalyzed by iron and vitamin B6, Communications Biology https://doi.org/10.1038/s42003-019-0431-5 ) Thus, it is important to emphasize this to show how non-specific AOAA acts. Conversely, it can also be used to emphasize AOAA can be used to block all enzymatic and non-enzymatic de novo H2S productions.
